# Seamless Navigation, 3D Reconstruction, Thermographic and Semantic Mapping for Building Inspection

**DOI:** 10.3390/s22134745

**Published:** 2022-06-23

**Authors:** Adrian Schischmanow, Dennis Dahlke, Dirk Baumbach, Ines Ernst, Magdalena Linkiewicz

**Affiliations:** Institute of Optical Sensor Systems, German Aerospace Center (DLR), 12489 Berlin, Germany; dennis.dahlke@dlr.de (D.D.); dirk.baumbach@dlr.de (D.B.); ines.ernst@dlr.de (I.E.); magdalena.linkiewicz@dlr.de (M.L.)

**Keywords:** absolute referencing, building inspection, building information model (BIM), multi-sensor data fusion, pixel co-registration, real-time self-localization and mapping, semantic model, seamless navigation, trifocal geometrical camera calibration, visual aided inertial navigation, visual odometry, 3D thermal mapping

## Abstract

We present a workflow for seamless real-time navigation and 3D thermal mapping in combined indoor and outdoor environments in a global reference frame. The automated workflow and partly real-time capabilities are of special interest for inspection tasks and also for other time-critical applications. We use a hand-held integrated positioning system (IPS), which is a real-time capable visual-aided inertial navigation technology, and augment it with an additional passive thermal infrared camera and global referencing capabilities. The global reference is realized through surveyed optical markers (AprilTags). Due to the sensor data’s fusion of the stereo camera and the thermal images, the resulting georeferenced 3D point cloud is enriched with thermal intensity values. A challenging calibration approach is used to geometrically calibrate and pixel-co-register the trifocal camera system. By fusing the terrestrial dataset with additional geographic information from an unmanned aerial vehicle, we gain a complete building hull point cloud and automatically reconstruct a semantic 3D model. A single-family house with surroundings in the village of Morschenich near the city of Jülich (German federal state North Rhine-Westphalia) was used as a test site to demonstrate our workflow. The presented work is a step towards automated building information modeling.

## 1. Introduction

Self-localization and environmental mapping are essential for a number of inspection and survey applications. Navigation and environmental mapping are fundamental components in this context and belong together.

Currently, the existing infrastructure-based self-localization services are designed/optimized for either indoor or outdoor utilization. They have several restrictions in accuracy and availability and are, therefore, not optimal for seamless navigation between indoor and outdoor spaces [1].

The global navigation satellite system (GNSS) is the universal outdoor localization system and provides up to centimeter accuracy under good conditions. In urban areas and indoors, accuracy and reliability degrade drastically. The influence of disturbed GNSS signals have been investigated for the last two decades by different research groups, e.g., Kuusniemi and Lachapelle [2], Ghinamo et al. [3] and many others. Nowadays High-Sensitivity GPS Receivers still lack satisfying indoor self-localization accuracy and reliability and are far from meter or submeter accuracy [4]. Moreover, GNSS Signal repeaters can not reach stable submeter accuracies, and, in addition, they have to be placed inside a building [5]. GNSS solutions are thereby not (yet) the first choice for indoor navigation and inspection tasks.

Vice versa, the enormous effort required for the installation and maintenance of indoor use-optimized radio-frequency self-localization services, such as WiFi or ultra-wideband (UWB), makes their expansion from indoor to bigger outdoor spaces difficult. Furthermore, radio-frequency-based self-localization techniques are not able to sense and hence map, the environment alone.

All these principle drawbacks have hindered operational seamless indoor and outdoor self-localization.

Passive- (cameras) or active- (e.g., LiDAR—light detection and ranging) based optical sensor systems and corresponding computer vision approaches can overcome these issues. Driven by robotics research, appropriate optical sensor-based self-localization and mapping solutions have become available and popular in recent years. Most prominent are Intels^©^, RealSense™ [6], Googles ARCore [7], and Microsoft’s HoloLense [8]; there are also others. These technologies are capable of self-localizing and mapping the environment indoors and outdoors simultaneously but still have partial limitations, e.g., accuracy and reliability over operation time in outdoor areas and missing global referencing. Such low-cost commercial self-localizing and mapping systems also lack additional information layers, such as thermal infrared (TIR) or others that might be useful or mandatory for particular inspection applications.

These aspects are a prerequisite for many inspection applications, where real-time self-localization and mapping capabilities within an absolute reference system and fusion with other geographic information are essential.

In this paper, we present an approach that overcomes all of the above-mentioned limitations. The overall objective of this research was to span an appropriate workflow for seamless real-time self-localization with an integrated positioning system (IPS) [9] and combine it with thermal 3D mapping for practical building/infrastructure inspection tasks. Thereby, it shall be possible to seamlessly navigate between indoor and outdoor spaces and to quickly survey critical infrastructures with adequate accuracy. Areas that are difficult to access or non-reachable with the IPS, e.g., a building’s roof, shall be complementarily covered by UAV data. The generated geodata products (operator’s positions over time/trajectory, thermal textured point cloud and building vector model) shall be available in the global satellite system (GPS) reference frame. The automated workflow shall be, in part, real-time capable and shall shorten the data processing time. This article is structured as follows: Section 2 reviews the related work; Section 3 describes the preliminary work and our experimental setup; Section 4 presents the methodology and the corresponding results when applied to our test scene data; in Section 5, our results are evaluated and discussed; Section 6 summarizes our conclusions and gives a future outlook.

## 2. Related Work

The following subsections give, without claim of completeness, a short introduction to related work in the particular research areas of self-localization, 3D reconstruction, thermal mapping, and functional building reconstruction from optical sensor data.

### 2.1. Self-Localization

Seamless navigation between indoor and outdoor spaces is a challenging task that comes with several difficulties that self-localization algorithms have to deal with. Examples of these challenges include spatio-temporal unavailability or disturbances of infrastructure-based self-localization radio frequency sensor signals, as they have been already mentioned for GNSS, or the transition between different self-localization technologies with different characteristics (sensing principle, data type, quality, time base, spatial and time resolution, coordinate system, range of availability). Solutions that are not dependent on any infrastructure are also affected. In the case of passive optical sensors, for example, difficulties arise, e.g., from rapidly changing light conditions in the transition between indoor and outdoor, resulting in short-term over- or underexposed images, which might cause problems within the feature point extraction. These are experiences the authors experienced in theie daily work with passive optical camera systems. Another example is the limited suitability of near-infrared pattern-based self-localization sensors outdoors. It seems that these difficulties may be adequately dealt with for combined indoor–outdoor environments via multi-sensor self-localization approaches by means of combining infrastructure-independent with infrastructure-based aiding components and additional a priori knowledge, e.g., maps. From the user’s perspective, operational self-localizing systems should be as small, lightweight, and as user-friendly as possible while at the same time being independent of environmental conditions and very robust and precise.

Smartphone-based systems have the advantage of many different sensors already being integrated into their design. In [10], the inertial sensor system of a smartphone is aided by GNSS when available, the direction of travel by a magnetic sensor, and in indoor areas with the help of a digital map. The transition from outside to inside is determined by the smartphone’s magnetic and light sensor. Le et al. [11] show how a person first navigates through a city with their smartphone in a vehicle and switches to pedestrian navigation when entering a building. For outdoors, an ORB-SLAM (feature-based simultaneous localization and mapping) solution is calculated from visual measurements and scaled by a GPS signal. The resulting solution is further fused with an inertial measurement unit (IMU) and GPS measurements using an extended Kalman filter [12]. An IMU-based pedestrian dead reckoning (PDR)/ORB-SLAM integrated system is used for indoor navigation.

Foot-mounted PDR systems apply a zero-velocity update when the pedestrian is not moving. However, in [13], it was shown that the analysis of differences between time-shifted measurements of acceleration and magnetic sensors can also contribute to improving the navigation solution. Additionally, a GPS receiver is intended for outdoor use, but it can also detect the transition from outside to inside. Furthermore, Peltola et al. [14] improve indoor position determination with the use of an ultra-wideband range measurement and an anchor-based Bluetooth fingerprinting system. A novel dual-mode filter design is used to fuse all sensor measurements. Map information helps to automatically select the most suitable filter for a given situation. Kourogi et al. [15] apply dead reckoning with body-worn, self-contained sensors, such as accelerometers, gyroscopes, and magnetometers, in combination with an active radio frequency identification marker system for indoor environments. Outdoor GPS helps to correct navigation errors.

There are also a number of hand-held systems or hybrids, such as a foot-mounted IMU. With the hand-held platform in [16], an extended Kalman filter fuses data from an ultra-wideband indoor positioning system with a classic inertial navigation system (INS)/GNSS. A total georeferenced station, which can track a hand-held system using a prism, serves as a reference for a multi-sensor system indoors and outdoors. The foot-mounted PDR in [17] is enhanced by a visual gyroscope/odometer obtained from monocular camera images. In addition, the vertical position is aided via sonar and a barometer. All measurements are integrated through a particle filter. The ground truth is provided by a commercial INS/GNSS solution.

### 2.2. 3D Reconstruction and Thermal Mapping

Three-dimensional thermal mapping has great potential in various applications such as building energy efficiency monitoring [18,19] and general object detection [20], as well as critical applications such as hotspot and fire detection. Fritsche et al. [21], for example, describe a LiDAR, radar (radio detection and ranging), and thermal imaging-based system to detect hazards that are potentially harmful to robots or firefighters, while Rosu et al. [22] show how to detect and localize heat sources in RGB and infrared (IR) textured meshes from pre-registered LiDAR scans.

Methods for mapping thermal information onto previously generated 3D point clouds have frequently been proposed in the last decade. If a geometrically calibrated sensor setup is used and the relative pose of the thermal camera to the 3D point cloud can be estimated accurately, the pixels of all cameras or light beams from LiDAR systems are co-registered with the thermal camera pixels, and the corresponding thermal pixel values can be assigned to the 3D points.

LiDAR allows for accurate and dense 3D reconstruction. Therefore, it is used to capture 3D base models by collecting and registering several laser scans acquired at different positions, as in [23] or [24]. Lagüela et al. [25] describe a large-scale vehicle-based application in which they generate a 3D point cloud by fusing LiDAR, GPS, and IMU sensor information. Högner et al. [26] connect and register four different devices (3D laser scanner, 2D laser rangefinder, stereo camera, and thermal camera) with the goal of using this synchronized data for the indoor thermal mapping of point clouds. Alternatively, 3D point clouds can be generated from passive optical sensors working in the visible range of light (VIS). Landmann et al. [27] describe a stationary but high-speed system that maps TIR data continuously onto a moving 3D model generated with a structured light approach. A hand-held system that uses a more lightweight RGB-D sensor for 3D reconstruction and works in a smaller indoor scenario in real-time is presented by Vidas et al. [28] and Müller et al. [29]. They apply a combination of iterative closest point (ICP) and video-based pose estimation to the thermal image stream. The most likely TIR information from several sensor images is assigned to the 3D points, for which a voxel-based occlusion test and a confidence value are evaluated. At the expense of the real-time capability, Schramm et al. [30] improve the robustness of the 3D thermal imaging system’s self-localization ability by adding an additional stereo camera. Visual odometry (VO) and simultaneous localization and mapping (vSLAM) is widely used for real-time outdoor applications. Irmisch et al. [31] used a hand-held stereo system with an IR sensor in a harsh and dynamic outdoor environment for large-scale direct 3D reconstruction and thermal mapping in global coordinates. Yamaguchi et al. [32] used a monocular RGB visual odometry approach to 3D reconstruction that involved superimposing a thermal image. The scale of the model was restored again using generated depth images from the VIS and IR domains, as well as the known fixed camera transformation of the combined system. Very precise 3D reconstruction results can be achieved with a structure from motion (SfM) approach, but only with considerable computational effort. Troung et al. [33] align a 3D model based on RGB images and a model based on thermal sensor data with scale normalization. They estimate the metric scale of the model based on the fixed-camera transformation between the RGB and thermal sensors. Patrucco et al. [34] combine aerial images from a UAV flight campaign involving multiple sensors with thermal textured models with spatial resolution from RGB images.

### 2.3. Building Reconstruction

Detecting façade planes and building footprints requires approaches that ideally employ terrestrial sensor constellations or airborne oblique imagery. Due to the steep viewing angles, façades are inherently poorly visible in traditional vertical aerial data products. Hammoudi et al. [35] proposed a building footprint extraction method based on terrestrial LiDAR data using a projection and the subsequent Hough transformation of the point cloud.

Techniques for the automated extraction of roof parts and their topology with a priori knowledge about building footprints described in the literature can be separated into two groups. The first group involves model-driven approaches. An example of an advanced commercial solution that uses the model-driven approach is the virtual city system [36], which can be applied to large datasets [37]. The second group consists of methods with a local approach. Parts of a roof area that are smaller than the plane they belong to are identified separately. Adjacent parts with similar properties are combined in the encircling roof area. This local approach was used by Sampath and Shan [38]. They initialize small clusters (roughly 500 points) within the point cloud. By analyzing the normal vectors of the initialized clusters, similar adjacent clusters are identified and merged as the roof area. A further example of the local approach was presented by Peternell and Steiner [39]. However, the calculation of the roof area is carried out by comparing planar neighborhoods between cells belonging to the 2D mesh.

## 3. Preliminary Work and Experimental Setup

The generation of appropriate real sensor datasets within a field measurement campaign required preliminary work that fulfilled a number of prerequisites. These include adaptions to the IPS hardware and operation software, calibration matters, and the preparation of the test site. These issues are presented in the subsequent subsections.

### 3.1. Trifocal IPS

In its basic configuration, IPS is a real-time-capable technology developed at the German Aerospace Center by the Institute of Optical Sensor Systems. IPS determines its position and orientation, the six spatial degrees of freedom (DoF), by means of a visual-aided inertial navigation approach [40] in a relative coordinate reference frame. Infrastructure such as global navigation satellite systems (GPS, Galileo, etc.) or WiFi is not required natively [9]. The IPS sensors include a stereo configuration of a robust industrial-grade monochromatic one-mega pixel camera [41] with global shutter mode and a tactical grade Analog Devices IMU [42]. The synchronization of the stereo images (10 Hz), the IMU (410 Hz), and any other self-localization and inspection sensors are handled by a field-programmable gate array (FPGA). The data capturing and processing is conducted by the IPS navigation and 3D reconstruction software. The output frequency of the IPS self-location can be changed as required. By default, it corresponds with the visual camera’s frame rate.

Dense depth map reconstruction from stereo imagery and further processing to point clouds can also be conducted simultaneously with self-localization. This allows for instant inspection results at speed and reasonable accuracy and gives high operational values to the operator. If it is required by the application, navigation and 3D reconstruction results can be further improved by post-processing, e.g., by bundle block adjustment.

IPS localization and 3D reconstruction capability and robustness have been evaluated in various environments in [9,43]. It is suitable for large-scale 3D reconstruction, as is required, for example, in mining inspection [44], as well as in other inspection tasks in partially tight or generally difficult environments such as tunnels, ships, and building interiors. In these scenarios, an operator does not necessarily return to previously visited areas. Other self-localization and mapping approaches that use feature catalogs for re-localization cannot take advantage of such application scenarios and also cannot compensate for the sensor data drift.

To gather thermal information, we extended the IPS using a thermal camera [45]. The trifocal camera setup (Figure 1) used in this approach, therefore, operates in the visible and long-wave infrared (LWIR) spectra (Table 1). The monochromatic stereo camera pair (VIS) is located on the front of the device with intermediate lighting units. The thermal camera is mounted on the IPS sensor head. It captures images in rolling shutter mode and performs frequent offset calibration during recordings, which makes the camera blind for a short period.

The LWIR camera stores thermal pictures via the Optris PIX Connect software [46] when manually triggered by an external push-button release on the handlebar. The trigger signals are time-stamped by the IPS FPGA, and the thermal images can thereby be synchronized with all other IPS sensor data. Due to the LWIR camera rolling shutter and offset calibration, thermal imaging was only triggered when the sensor movement was paused for a moment.

### 3.2. Calibration

An accurate camera system calibration is a precondition for trifocal sensor data fusion at the pixel level to acquire thermally-colored point clouds. In order to take all sensors, including the stereo- and thermal camera, as well as the IMU, into account, the calibration approach is divided into three parts. Initially, the stereo cameras are calibrated, and the left camera’s principal point serves as the origin of the system’s reference frame.

Secondly, the IMU’s spatial alignment is determined within this reference frame. These are laboratory calibration steps. Finally, the thermal infrared camera’s intrinsic parameters are determined and co-registered with the left stereo camera. This last step is carried out in the field just before the inspection measurement is run due to the thermally unstable LWIR optics. The following text passages describe the sensor calibration in more detail.

#### 3.2.1. Stereo Camera

In order to remove distortion effects from images, it is necessary to derive the geometrical calibration parameters. They are gathered under the term *intrinsic calibration* and include the focal length, principle point, and radial symmetric distortion. If several optical sensors are combined, an additional pre-requisite is knowledge about the spatial alignment between all sensors. These parameters are gathered under the term *extrinsic calibration* and are modeled by additional relative orientations ${\left(R,t\right)}_{c1}^{c2}$ and ${\left(R,t\right)}_{c1}^{c3}$, with a rotation (*R*) and translation (*t*) between the cameras $c_{1},c_{2}$ and $c_{3}$. For this purpose, a planar aluminum calibration chart (Figure 2a,d), also allowing for TIR camera geometrical calibration due to thermal emission pattern differences, is used to derive intrinsic and extrinsic parameters for the trifocal sensor setup at once. The two-step approach first estimates intrinsic and extrinsic parameters with a linear optimization [47]. Given these initial values, the non-linear least-squares problem of the distortion model is solved with the Gauss–Newton algorithm. For the automated detection of chessboard corners, as well as the bundle adjustment, the solution presented in Wohlfeil et al. [48] is used.

#### 3.2.2. Camera to IMU

After assembling the entire sensor system, the spatial 6 DoF between the left camera ($c_{1}$) and the IMU ($i_{1}$) are determined once. For this purpose, the 3D distances ($t_{c1}^{\;\;i1}$) between the two coordinate systems are measured manually or derived from a technical drawing. The angles of the rotation matrix $R_{c1}^{\;\;i1}$ are estimated using a static method [49] that requires several different poses of the IPS in front of a calibration chart. Thereby, directional information relating to the local tangential plane is derived from IMU accelerometer outputs. With the help of the measured camera orientation with respect to the calibration chart, an over-determined system of non-linear equations can be set up that is solved for the desired rotation.

#### 3.2.3. Thermal Camera

Thermal imaging sensors could be geometrically calibrated like conventional cameras, but calibration in the mid-wave to long-wave infrared spectrum poses several challenges. One is the relatively low number of image pixels, which demands a good fit of target size and camera field of view. Another challenge is the creation of good contrast for the features on the calibration target. Targets either containing self-emitting elements or reflecting ambient radiation are suitable. For the spatial alignment of visual and thermal imaging sensors, the stereo cameras have to recognize the target as well. As in Choinowski et al. [50], we have chosen an aluminum chessboard target, which is portable and has proven to work well in the visual spectrum. Compared to Choinowski et al. [50], our chessboard contains one geometric and one radiometric improvement. On the one hand, the size of the chessboard patches is increased from 36 cm${}^{2}$ to 72 cm${}^{2}$. Although this means roughly half the number of corner points on a comparable area, the detection of corners itself becomes much more robust. On the other hand, only black patches are printed on custom static cling, while the formerly white patches now remain blank aluminum. When positioned facing the sky, the blank chessboard parts are now comparable to a mirror with high reflectance in the LWIR range. In this way, the thermal gradient between the sky and the ambient temperature, combined with the different emissivities of the printed and blank patterns, gives a high contrast to the chessboard. The formerly faint contrast is enhanced from 1 to 10 K or more. Corner detection, in particular, benefits from these improvements since it relies on good contrast and well-defined edges along the pattern. An exemplary calibration image triplet is shown in Figure 2. A setup for the calibration and spatial alignment of multi-camera systems with planar reference targets can be found in Luhmann et al. [51]. Corner detection and a subsequent bundle adjustment is conducted on all synchronized calibration image triplets again using the solution presented in Wohlfeil et al. [48]. The chessboard pattern is captured in 28 different poses under the open sky. Note that the different poses present a balanced set, with several rotations and distances to de-correlate intrinsic and extrinsic calibration parameters as much as possible. All the remaining re-projection errors are in the sub-pixel range, as shown in Figure 3, enabling the system to generate metric point clouds colored with thermal intensities in real-time. The calibration results of the intrinsic and extrinsic parameters for the trifocal sensor can be found in Table 2.

### 3.3. Ground Control Point Survey

Overall, seven ground control points (GCP) are surveyed with a real-time kinematic-capable Leica GPS receiver [52], as shown in Figure 4b. The mean horizontal accuracy is 1.0 cm, while the height quality is, on average, 1.3 cm. They are distributed around and near the study building. To guarantee precise angular accuracy for the conducted IPS inspection runs, the remaining two points span a rectangle roughly 40 m further north. Each reading point is equipped with an AprilTag, which can be recognized automatically by the IPS during inspection runs. Furthermore, the survey is conducted prior to the UAV flight campaign so that the marked GCP can also be used for the data fusion between terrestrial and aerial data products.

### 3.4. Test Site and Field Experiment

The test site is an abandoned single-family house (9 × 9 m) with two floors and a basement embedded in a rural environment in the German village of Morschenich, which is located within the North Rhine-Westphalia open-pit coal mine territory, not far from the city of Jülich. The building has been used for research activities by the German Aerospace Center’s and Jülich’s solar research institutes in the past and provides a validated test ground for remote sensing test and validation measurement campaigns. It is freely accessible from all directions, and the area also allows for the possibility of flying UAVs around the building without obstacles. During the very sunny day on which measurements were taken, there were different light conditions, which put a lot of demands on the exposure control of the visible cameras. For example, transitioning from outside to inside of the house, or from very bright to dark, can be challenging as the cameras need time to adjust exposure control. There were no additional light sources in the house itself. Very dark rooms without windows or daylight were inspected, as well as very bright passages with direct sunlight coming through the windows. Artificial heat sources were placed in the house beforehand. We also took non-contact infrared thermometer measurements, which serve as contrast sources and references for the thermal imaging data.

### 3.5. Aerial Imagery

Since the IPS measurements are bound to terrestrial standpoints around and inside the building, aerial imagery was used to map the roof structures as well. The flight was conducted using a DJI Mavic Pro [53] and the flight planning software Pix4DCapture [54]. Three stripes with 14 images each were captured at 40 m above ground level, leading to a ground sampling distance of 1.5 cm. The UAV camera is not calibrated since it works only in auto-focus mode. In combination with the Mavic’s consumer-grade GNSS receiver, the aerial images of the UAV cannot be used for direct georeferencing. Five ground control points (Figure 4a) are visible in the image mosaic and were used for precise georeferencing.

## 4. Workflow and Results

Our process chain, as shown in Figure 5, consists of three thematic blocks. The whole process pipeline can be executed, in principle, within one day for a small size single-family house. The needed time for the workflow scales with the measurement volume, the building size and complexity.

First, a georeferenced trajectory, covering the whole in- and outside area to be inspected, is generated at run-time during the inspection walk with the calibrated IPS over DGPS (differential global positioning system) surveying the ground control points. The detailed working steps of the *IPS self-localization* block are described in Section 4.1.

With the estimated global 6 DoF, a global referenced point cloud is calculated. For this purpose, dense depth maps are computed from the IPS stereo image pairs, transformed to point clouds with global coordinates, and subsequently aggregated. This resulting georeferenced point cloud covers the local surroundings with up to eight-meter object distance along the inspection path. The 3D reconstruction from subsequent stereo image pairs and point cloud aggregation was carried out in post-processing in this case, but the approach is principally run-time capable. If the application requires it, the IPS point cloud geometry can optionally be further improved by bundle adjustment in post-processing, but this was not part of the presented work. Knowing the intrinsic and extrinsic parameters of the IR sensor and the IPS stereo camera, which is determined by trifocal geometrical calibration as described before, the thermal information is mapped in post-processing onto the point cloud in a false color representation. Section 4.2.1 describes the steps of this 3D and thermal mapping in detail. The second part of the *Thermal 3D reconstruction* block, aero triangulation, and multi-view matching of UAV-derived images, which is also performed in post-processing, is described in Section 4.2.2.

The third thematic block, *building model reconstruction*, in Section 4.3 describes methods for the building footprint and façade extraction from a point cloud, which is predominantly generated by the terrestrial IPS and partially from aerial UAV imagery and without a priori knowledge, such as cadaster information. The method for segmentation used in this paper is a data-driven approach. In particular, façades and roof planes are reconstructed through linear regression, region growing, intersection evaluation, and topology analysis.

### 4.1. IPS Self-Localization

In order to minimize the influence of temperature-dependent effects in the electronic sensor components on the measurement data, the IPS is supplied with power for a certain period of time at the beginning of each measurement campaign but without an active measurement task. The measurement runs in Morschenich started in the outdoor area around the single-family house. At the beginning of each run, a static and dynamic procedure is carried out that is used to initialize the system states. During a predefined phase of no movement, the IMU’s gyroscope offsets can be calculated. In the dynamic part, the IPS is moved slowly around all three spatial axes. As a result, the accelerometer offsets are correctly estimated or separated from the two horizontal angles of the initial system orientation. After this short initialization phase, IPS is placed on the floor, the estimated state vector is reset, except for the IMU offsets, and the system is ready for use.

#### 4.1.1. Local Trajectory

First, the multi-sensor fusion calculates a navigation solution in a local tangential system, which is spanned by the initial alignment of the IMU. The horizontal angles of the rotation are determined by the orientation of the acceleration sensors to the Earth’s gravity field. The vertical rotation angle and the 3D starting position are freely selectable and are set to zero.

The *Inertial Navigation System (INS)* in Figure 5 consists of a prediction and a correction part. The so-called strapdown algorithm [55] takes over the summing up of the IMU’s accelerations and angular velocities to the resulting system pose. The integral solution is insufficiently accurate due to faulty sensor measurements, e.g., caused by noise or offsets, but also modeling errors in the kinematic equations. By choosing suitable additional sensors, these errors can be estimated using an extended Kalman filter [12] and corrected iteratively. In the IPS basic configuration [40], a stereo camera system is included for this purpose, from whose temporally consecutive image pairs visual odometry (3D relative orientation and position) can be calculated. At a time step, natural landmarks are first extracted in the left camera image, which is then found again in the right camera image. These stereo features are triangulated and then tracked in the image pair of the next time step. Thus, the corresponding 6 DoF relative change in the movement of the stereo camera pair can be estimated using a solver for a set of non-linear equations, including random sample consensus (RANSAC).

#### 4.1.2. Georeferenced Optical Ground Control Points

To switch from local to global navigation, georeferenced ground control points are required, which are precisely measured in advance of the actual measurement run using a precise mobile DGPS receiver (Section 3.3). These global references were distributed in the area around the single-family house in such a way that very good GPS reception quality could be guaranteed.

The assignment of global coordinates to the IPS positions takes place by means of the optical sensors, where AprilTag targets [56] can be detected in the stereo image pairs. By placing the targets on the previously measured DGPS ground control points, the respective AprilTag identification (ID) is assigned to exactly one geographic coordinate. The now-global optical landmarks, marked by green rectangles in Figure 4a, can be automatically recognized in the subsequent measurement run with the help of an OpenCV detection algorithm [57]. The distance between the fixed AprilTag target on the ground and the left camera of the moving IPS is calculated by triangulation and transformed into the IPS body system using a fixed alignment matrix (Section 3.2.2).

#### 4.1.3. Switch Logic

When passing the georeferenced AprilTags, they are automatically recognized in both camera images, which prevents the IPS from explicitly having to be placed on the ground. Internally, the triangulated distance between the left camera and the ground, the associated geographic reference position (GCP), and the local IPS pose are assigned to the AprilTag ID. As soon as an AprilTag is detected, this extended measurement is fed to the input selection logic in Algorithm 1.
**Algorithm 1** Selection logic for switching from local to global navigation**Data**   AprilTag triangulation reading, local IPS pose, global GPS reference coordinate**Result** 6 DoF homogenous transformation & associated covariance matrix
1:append ← true2:**if** current tag ID ∈ ID list **then**3:    **if** current tag ID = last item in ID list **then**4:        **if** current tag Euclidean distance > Euclidean distance of last item in ID list **then**5:           append ← false6:        **end if**7:    **end if**8:**end if**9:**if** append **then**10:    ID list ← add current tag ID11:    local point list ←  calculate local position from IPS pose & AprilTag triangulation12:    global point list ← calculate ECEF position from GPS reference13:**end if**14:**if** number of unique IDs in ID list ≥ minimum ID list size **then**15:    (transformation, covariance) ← 6 DoF estimation from local & global point lists16:    valid trafo ← true17:    **for each** item in local point list **do**18:        valid trafo ← residual among global & transformed local points < max. residual19:    **end for**20:    **if** valid trafo **then**21:        **return** (transformation, covariance)22:    **end if**23:**end if**


As an input test, the current AprilTag ID is compared with an ID list of tags that have already been saved. In the case of a new ID, the extended measurement should be appended to the corresponding list (lines 10–12). If the ID already exists, a further test checks whether the current and previous IDs are the same (line 3). If this test is also passed, an exclusion criterion involves checking the Euclidean distances of the current and previous triangulation measurement (line 4). Only triangulated measurements that come closer to the AprilTag over time may be added. If the aforementioned tests are passed or when a new ID is found, the current extended measurement should be appended to the corresponding list (lines 10–12). In addition, a local position-measuring vector is formed from the triangulated camera–AprilTag distance and the current IPS pose, which is added to the local point list. The associated global GPS reference is converted into Cartesian coordinates and forms a new measurement of the global point list. If both point lists reach a certain minimum number of 3D vectors (line 14), the homogeneous transformation matrix, including the associated covariance matrix, is estimated, which can convert the local navigation solution into the global coordinate system. The transformation matrix represents three parameters of rotation, as well as translation and a scaling factor. At least three pairs of 3D vectors in local and global coordinates are required to estimate these seven unknown values. The estimation vector of the non-linear system of equations is initialized with a static seven-parameter transformation method so that only a few iterations are required to calculate the solution in the following Gauss–Newton approach. The output test now consists of calculating the residuals for each input point set. If all list pair residuals fall below a defined maximum residual value (line 18), the transformation matrix and its covariance can be output as a valid solution, and the switchover from local to global navigation can be carried out.

#### 4.1.4. Global Trajectory

Our global target frame is the Earth-centered, Earth-fixed (ECEF) coordinate system; the origin is located in the center of the modeled Earth ellipsoid [58]. Its *X*-axis is defined as the intersection line between the planes of the equator and Prime Meridian. The *Z*-axis is also the rotation axis of the Earth, and the *Y*-axis results from a 90 degree angle to the two previously mentioned axes. Since all IPS sensor measurements and external ground control points deliver Cartesian values and subsequent processing modules in the process chain require Cartesian trajectory values, we chose this global coordinate system.

After successfully switching to global navigation mode, IPS is now able to thermally and geographically reference the interior of the single-family house. In addition, new or recurring AprilTag targets can still be used as absolute support information, thus improving the navigation solution. An optimized trajectory, shown as red dots in Figure 4a and Figure 6, is used for the following processing steps, which is automatically calculated at the end of an inspection run using a so-called fixed-interval smoothing approach [59]. For this purpose, all system states and covariances of the Kalman filter must be stored in the forward branch during the data fusion, from which the smoothed trajectory is then calculated in a subsequent backward recursion. The advantage of this smoothed trajectory is that both past and future measurements are included in the calculations, which makes the solution more accurate than the estimation of the filtering in the forward branch. However, to use a fixed-interval smoothing algorithm, all measurements of the test run must be available.

### 4.2. Thermal 3D Reconstruction

The next section describes the 3D reconstruction and thermal mapping of the trifocal sensor data to point clouds with the thermal layer in the absolute reference frame.

#### 4.2.1. Trifocal Thermal 3D Mapping

The IPS, with its stereo camera approach, generates two images at the same time, which are used for reliable visual odometry estimation and for the generation of high-density depth maps by stereo matching. This computationally expensive processing step is implemented in OpenCL and executed on a graphics processor unit (GPU). After image rectification, a semi-global matching algorithm (SGM) [60] with a census cost function as described in [61] for the data term is used (Figure 7b).

The frame rate for point cloud generation is dynamically adjusted based on the IPS navigation solution. If a substantial difference in pose or time to the previously used image pair is reached, a new 3D point set is extracted from the next depth map and transformed into a local tangential Cartesian coordinate system derived from the ECEF starting point of the trajectory. This allows the following point cloud operations to be performed in an efficient data format. The connection of the resulting point cloud to ECEF is given by the inverse transformation. Subsequently, the point sets generated from single image pairs are accumulated into a high-density cloud and filtered into a voxel grid of an appropriate resolution (1 cm for the test data set) and size. In subsequent steps, these partial 3D point clouds can be aggregated to 3D models of the whole observed object. In Figure 6, multiple views of the whole 3D point cloud from the IPS image sequence are shown. The voxel generation and cloud filter steps are based on the Point Cloud Library [62], and all point cloud figures presented in this section are rendered with CloudCompare [63].

IPS can record the data of additional sensors synchronized with image and IMU data simultaneously. In the case of IR-image data, whose pixels are co-registered to the stereo camera pixels by trifocal geometrical calibration, they can be immediately mapped onto the point cloud. Figure 7a,c show a left IPS image and an IR image recorded at the same time. The RGB or panchromatic color values of the 3D points are blended with the thermal information in color representation from the additional thermal camera. The 3D object view helps us to understand the temperature distribution in the observed scene. Figure 8a shows a point cloud generated from one image pair, and Figure 8b show the corresponding point cloud with mapped temperature coding colors from the IR image.

Compared to the 10 Hz frame rate of the IPS panchromatic stereo cameras, only a few IR images have been recorded for the test data set. For a higher point cloud density that carries thermal information, 3D points can be collected from multiple image pairs, which have been recorded shortly after the IR image within a certain time window and with similar camera poses to the IR recording.

These additional 3D points are projected into the valid IR camera image. For this, the interior orientation and distortion parameters of the IR sensor, which were determined in the calibration process in Section 3.2.3, as well as the relative orientations of the IPS camera poses to the IR sensor at recording time, are taken into account. Then the color or measured temperature values can be assigned directly to the respective surface points.

Three-dimensional points must be excluded from incorrect IR color assignment if they were occluded by another object and could not be seen from the IR sensor at recording time. Considering the near real-time capability requirements for future applications, a voxel-based occlusion algorithm is applied, building on the implementation of [64] in [62]. Figure 8c demonstrates the improved point cloud density and coverage when a single IR image is mapped to a point cloud generated from an IPS image sequence. Some parts (e.g., behind the ceiling lamp) are hidden from the IR camera’s view and are, therefore, excluded from IR color mapping.

In a subsequent automatic filter process, voxels and their additional information are removed based on, e.g., the frequency of the additional camera images, the number of 3D points found per voxel, the ratio of the number of points with and without additional information in a voxel, and their reliability. Figure 9 depicts multiple views of the complete IR-mapped point cloud.

#### 4.2.2. Point Cloud Generation from Aerial Imagery

In order to derive a complete point cloud of a building and the surroundings, two different kinds of input datasets describing the scene from distinct but complementary perspectives are required. Firstly, we used the terrestrial IPS point cloud, which is described in the previous section. Secondly, we used a point cloud containing the roof structures. These structures are poorly seen from terrestrial IPS standpoints, but they have been imaged by a commercial UAV system [53] and post-processed to a point cloud.

The commercial software Pix4DMapper was used for the point cloud calculation from aerial imagery. The recorded UAV data set comprises 40 RGB images. Interior and exterior camera orientations are given in an exif metadata form and can be used directly in Pix4DMapper. The software performs aero triangulation and reconstructs the point cloud. Four ground control points have been used for georeferencing.

The IPS (Figure 6) and the aerial point cloud were merged together using the CloudCompare software and its ICP algorithm. Figure 10 shows the resulting point cloud where the outer building structure is completely 3D reconstructed.

### 4.3. Building Model Reconstruction

The subsequently described 3D building model reconstruction approach gives just an overview and is, therefore, very superficial and selective. For more details, refer to Frommholz et al. [65], Linkiewicz and Meißner [66], and Dahlke et al. [67]. In order to calculate a 3D vector model, as shown in Figure 11f, façade pieces are identified first. For this purpose, point cloud voxels get projected onto a planar 2D cell grid (Figure 11a), and the direction of the point cloud within all particular cells is obtained via RANSAC. The statistically evaluated point clouds of adjacent cells in the same direction are then grouped together. The grouped points (Figure 11b) are approximated by a regression line to vector segments. These are then intersected to obtain a closed contour (Figure 11c). The roof structure (Figure 11e) is treated in the same way as the façade described above but in 3D space. In addition, a breadth-first region growing on pre-selected grid cells is made (refer to [66]). This fills the gaps within the roof structure and ensures topological consistency. This provides a spatial relationship between the particular structural parts of the building.

In this context, it is the precondition for intersecting roof surfaces in the next step. Finally, the façades and roof planes, representing the building semantics, are intersected to form a finite building hull (Figure 11f).

Thus far, the roof outline always coincides with the building footprint when intersecting adjacent surfaces as described above, and overhangs are not preserved. By using the building footprint and its projection onto the digital surface model (DSM), the overhang is calculated with sub-pixel accuracy (refer to Dahlke et al. [67]). If the DSM contains discontinuities (step edges) in close proximity to façade edges, they are very likely describing the true outline of roofs, including their overhang. A line parallel to the façade edge sweeps outwards with a step size of one pixel. For each step, the median of the line’s height profile is stored. The sequence of the stored median values provides a (median) height profile perpendicular to the direction of the façade edge. A first approximation of the distance to the façade is the position with the highest gradient within this perpendicular profile. Around this first approximation, a cubic polynomial is fitted. The inflection point of this continuous function is used as a sub-pixel accurate position for the overhang boundary. The reconstructed roof polygons of the respective building model are then extended accordingly (Figure 12a). Due to the different normal vectors of roof and wall planar faces, the assignment of semantics is straightforward. All horizontal normal vectors define a wall face, while the remaining faces are part of the roof (Figure 12b).

## 5. Evaluation and Discussion

The following section evaluates and discusses the proposed approach. Challenges, which were induced by environmental conditions as well as measurement equipment-handling specifics during the field experiments, and their impact on the processing results are reflected. The local geometric accuracy of the IPS point cloud, and the 3D vector model is determined and compared with reference data. The section also evaluates the thermal infrared sensor temperature data with reference examples.

### 5.1. IPS Self-Localization

The outer shell and two floors of our test object were inspected and geographically referenced with IPS. In order to be able to create a valid 3D reconstruction of the building, the position and orientation of the localization system must be consistently estimated. Various challenges had to be overcome. For example, rapidly changing light conditions when moving from outside to inside or the lack of lighting in the house can be problematic for a navigation system that is based on processing stereo camera images. In addition, the wide range of spatial image sequences, such as narrow passages in the hall or stairways on the one hand, and rather extensive passages around the house on the other, can be demanding for image processing algorithms.

The overall accuracy, including the switching from local to global coordinates, is determined by the accuracies of the individual measurements used from the AprilTag detection, the local IPS trajectory, and the global ground control points. The GCPs measured previously using DGPS have a very high position quality. The uncertainties of the local IPS pose result from the fusion of all sensor inputs of the visual-inertial navigation system. Since image data is also used in addition to the IMU, the local IPS accuracy is also determined by the number of features seen in the environment. The seamless transition to a global or geographically referenced navigation solution was implemented with the help of previously measured optical markers. The AprilTag targets were detected by the cameras during run-time so that their absolute positions could be used to switch the self-localization to global coordinates. The quality of the automatic tag detection and processing is determined by the size of the tags seen in the image, which is influenced by the distance to and orientation of the camera [56]. It is expected that the localization accuracy will decrease with increasing distance. Furthermore, the angle difference between the optical axis of the camera and the normal vector of the tag is crucial. If the tag rotates out of the camera’s view, the accuracy and detection rate will decrease. In order to minimize both of these risks, only tags that were detected as close as possible to the camera are processed. In addition, the quality of the switching described in Algorithm 1 can be influenced by parameterizing the number of AprilTags used (n≥3), and the intended uncertainty for the estimated transformation matrix.

As a consequence, IPS is drifting over time in both indoor and outdoor environments, as it is the nature of infrastructure-free self-localization techniques. Therefore, the accumulated errors should be nearly the same indoors and outdoors. The error can just be reduced by external infrastructure such as optical markers or radio frequency self-localization techniques (WiFi, beacons, GPS, etc.). In the indoor/outdoor transition areas, the automatic camera exposure time setting can lead to over and underexposed images, so the visual odometry does not provide a navigation solution for a short time period. However, this is not supposed to be such a critical issue because the strap-down navigation provides valid results in a short time period and can bridge over temporal visual odometry non-availabilities.

No matter if there are surveyed GCPs used, which serve as a global reference (outdoors in our case) or they are just be used for some drift-compensations by means of re-localization on the AprilTags (indoors in our case), drift between the optical markers always exist. In conclusion of the drift discussion, accumulated drift errors cannot be corrected completely, which results in some displacements of the building structure, but the drift rate indoors and outdoors should be nearly the same. The amount of this displacement depends essentially on the building size, the density and distribution of optical markers, the texture of the investigated scene, and the moving profile during the inspection run.

### 5.2. 3D Reconstruction and Thermal Mapping

Dense depth map reconstruction from IPS stereo imagery by means of SGM and assembling them to point clouds using IPS 6 DoF poses enables the 3D mapping of combined and large indoor and outdoor environments in a global reference frame. The outer building hull is reconstructed without gaps and most of the inner hull. This was due to the complementing UAV and IPS from ground sensing. Although several optical markers (AprilTags) were used for referencing and re-localization out- and inside, the drift of the IPS navigation over time could not be prevented completely. This led to partly geometric instabilities in the building structure and shows up, e.g., in some double walls. Nevertheless, the local geometry stability of the IPS-generated point cloud is reasonable, as wall length measurements and a comparison with the building floor plan show (see Table 3 and Figure 13 and Figure 14).

Thermal mapping by means of pixel-wise temperature value assignment to point cloud voxels shows no visible displacements. IPS point cloud texturing with thermal sensor data works very well and is evidence of the high-quality trifocal geometrical sensor calibration processing pipeline. For a complete pixel-wise point cloud thermal layer texture, a higher thermal infrared image frequency and other thermal sensor imaging properties (a larger field of view and higher sensor pixel size) corresponding to those of the IPS stereo camera would be necessary.

All steps, including trajectory estimation, dense depth map generation in a sufficient frame rate, disparity resolution, and thermal mapping, can be carried out in real-time using a capable laptop. This provides the opportunity to generate and possibly view dense 3D point clouds enriched with thermal information during an ongoing measurement and for the entire area of interest. While the overall accuracy of the 3D model is highly dependent on the accuracy of the generated trajectory, the quality of the point cloud in detail is strongly influenced by the stereo camera parameters, especially the pixel resolution and base length. For a fixed camera setup, the depth resolution and local accuracy of 3D points are mostly determined by the minimum distance of the objects to the camera while passing them during the measurement run.

The indoor and outdoor environments can be acquired by the complete IR mapped point cloud combining IPS with a thermal camera and complementary UAV data. As there are certain drift-induced IPS self-localization inaccuracies, the building structure as a whole is showing some voxel displacements in parts of the point cloud as already described above. If at all, this could just be further reduced by subsequent and time-costly bundle block adjustments or by means of ICP processing, which have not been addressed within the presented work. Anyhow, it is plausible, for certain inspection applications, local measurement accuracies, e.g., on particular windows or walls, and their inclusion within a global digital building model with displacements in a certain range is already beneficial for inspectors. On a local level, the 3D length measurements show similar deviations for both environments to the reference, see Section 5.4.

Finally, from a business model view, it is always a trade-off between the application accuracy needs and the available resources (money, personnel, time) to generate and provide the digital data to end-users. In which situations the proposed approach might come to limitations or is not suitable for a specific application task is generally not easy to say. This needs to be incrementally evaluated by future work in real inspection scenarios on larger buildings/infrastructure together with end-users and industry partners.

### 5.3. Building Model Reconstruction

The shown building reconstruction method works automatically and is stable. However, it is to be noted that the accuracy of the 3D vector model strongly depends on the quality of the point cloud. The noisier the point cloud is, the larger the grid for the local regression needs to be. It follows that if elements of facades and roofs are smaller or equal to the grid, they cannot be reconstructed. This dependence has to be considered in the context of future work. For the reconstruction of the interiors of buildings, which are often more complex and more filigree than the outer shells, the input point cloud has to be almost noiseless.

### 5.4. 3D Accuracy

To evaluate the accuracy of the calculated IPS point cloud, some horizontal segments were measured (Figure 14). The measurements were compared against the reference data (Figure 13). Reflecting the considerable residual noise from SGM and the inherent simplification errors, an average difference of 0.07 m indoors and 0.03 m outdoors has been found between the reference and the IPS point cloud values. The results of the selected distance measurements are shown in Table 3.

### 5.5. IR-Texture Temperatures

Four pots with heated water were placed inside the building. A thermal image with three of these pots can be seen in Figure 15a,c. A non-contact infrared thermometer (Figure 15d) was used to verify the water and pot temperatures during an inspection run. Since moving persons in the room could have disturbed the IPS self-localization accuracy and could have occluded the water pots while image recording, the thermometer measurements were taken approximately two minutes in advance. In total, the temperatures of four water surfaces and two pot walls were measured with an infrared thermometer and the thermal camera as a reference (Table 4).

The previously mentioned two-minute time gap is the main reason why almost all reference thermometer values are slightly warmer than the subsequent IPS thermal image recordings, as the water had cooled down a bit in the meantime. Another observed effect is a broader temperature range on the pot walls compared to the more homogeneous temperatures on the water surfaces.

## 6. Outlook

The proposed workflow is intended to be further consolidated. On the one hand, the goal is to derive directly georeferenced, dense and thermal textured point clouds with veritable ad-hoc geometric stability and high spatial coverage in combined indoor/outdoor environments at run-time during an inspection walk with the IPS and make the approach thereby even more attractive for inspection applications. Therefore, 3D reconstruction and thermal mapping shall be integrated into the IPS real-time capable workflow, and the self-localization shall be further improved, e.g., by catalog matching and/or appropriate artificial intelligence methods. On the other hand, a thermal layer shall also be included in aerial mapping to attain complete thermal-textured point clouds of inspected objects with a big vertical extent. This is required as the hand-held, and helmet IPS inspection systems have a small stereo camera basis and are limited to near-3D reconstruction up to a maximum of an eight-meter object distance. Thereby, certain areas, such as building roofs or parts of them, cannot be surveyed manually from a further distance with the IPS. That is why aerial thermal 3D mapping would complement the inspection in a meaningful way.

For many applications, such as BIM modeling or disaster management, not only the outer shell of a building but also the interior needs to be reconstructed. Responding to this demand, the developed method for outer shell building reconstruction will be adapted to the modeling of interiors. For this purpose, algorithms will be further improved, and new methods are considered to be embedded within the workflow. Furthermore, we plan to fuse the 3D vector model data and the thermal information. To enable this connection, the thermal information will be projected onto 3D vector models. Thermal-textured 3D vector models will form the basis for further energetic analyses.

Further improvements on the sensor side shall also be considered. In order to use optics with a broader field of view for the thermal camera while still being able to automatically detect the calibration pattern, we suggest at least a fourfold resolution increase compared to the Optris PI 450 (Table 1). The integration of further inspection sensors, e.g., acoustic camera, LiDAR, radar, and gas sensor, shall also be investigated. Thereby different thematic information layers can be generated and assigned to trajectories, point clouds, and 3D vector models to enrich BIM.

The manual inspection of large buildings and critical infrastructures could be sped up using the proposed workflow and give BIM added values. This could enhance evaluation capabilities and action prioritization for several inspection applications, e.g., energetic building restoration and maintenance and disaster management, and may make a contribution towards higher efficiency and better decision making in these areas.

## Figures and Tables

**Figure 1 sensors-22-04745-f001:**
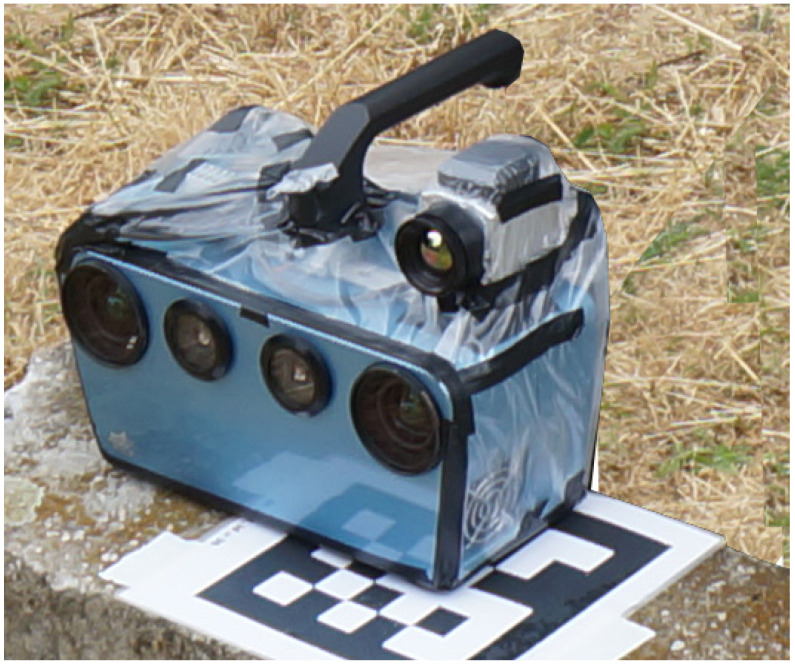
Trifocal IPS with a thermal (on top) and two visual cameras (front).

**Figure 2 sensors-22-04745-f002:**
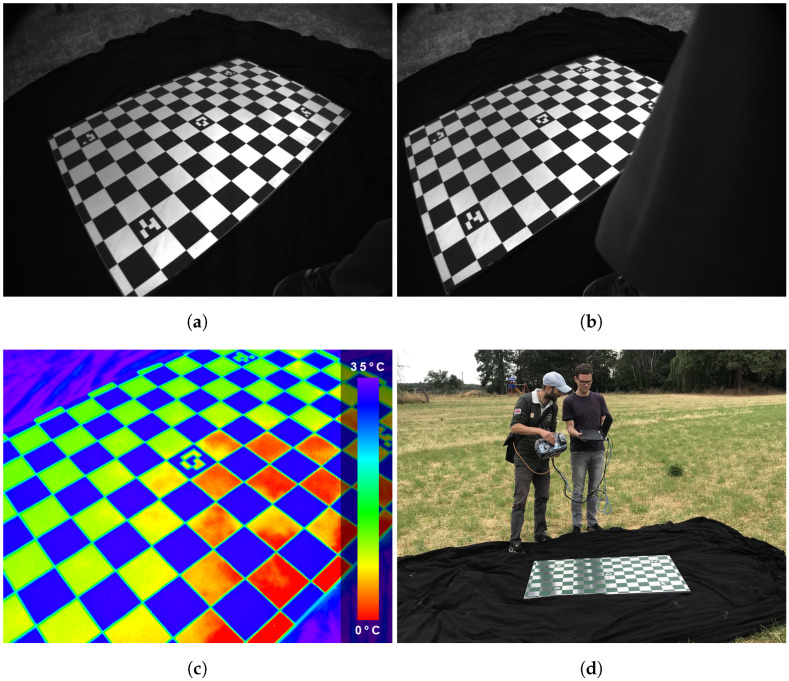
Trifocal geometrical chessboard calibration in the field just before an inspection run: (**a**) left IPS camera; (**b**) right IPS camera; (**c**) color-coded thermal image (color bar with temperature values); (**d**) image acquisition.

**Figure 3 sensors-22-04745-f003:**
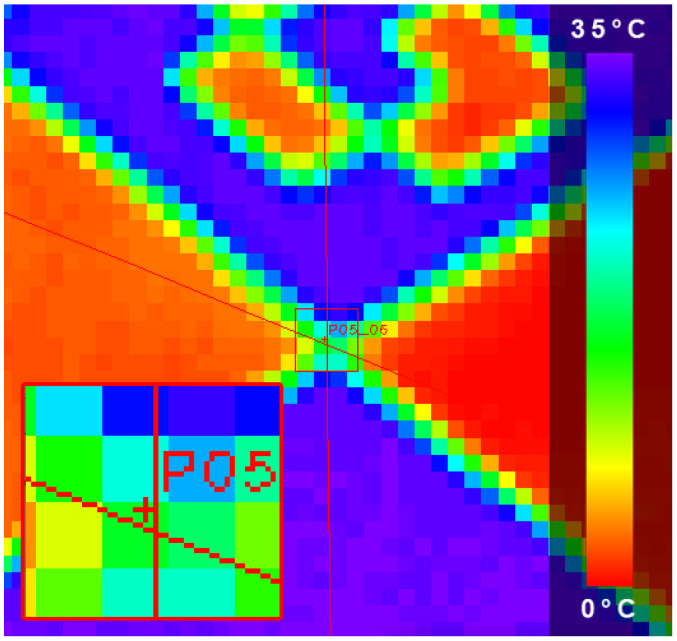
Epipolar lines at a selected point (color bar with temperature values).

**Figure 4 sensors-22-04745-f004:**
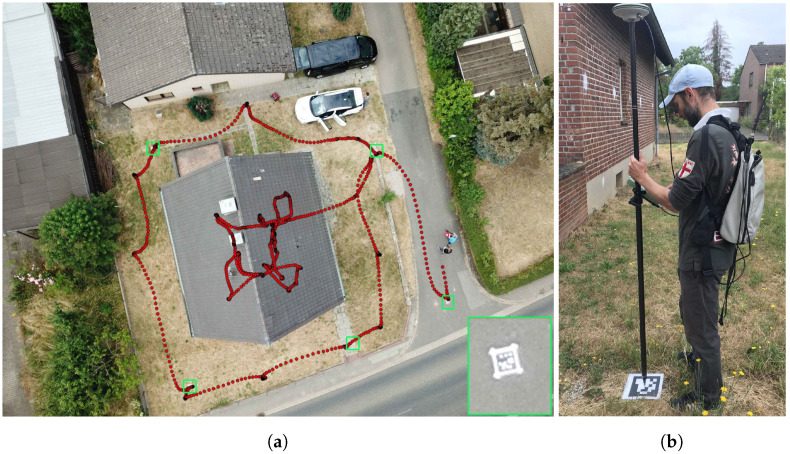
(**a**) One of 42 aerial images overlaid with DGPS georeferenced AprilTag targets showing GCPs (green rectangles) and a section of the IPS trajectory (red dots). (**b**) Leica 1200 DGPS surveying.

**Figure 5 sensors-22-04745-f005:**
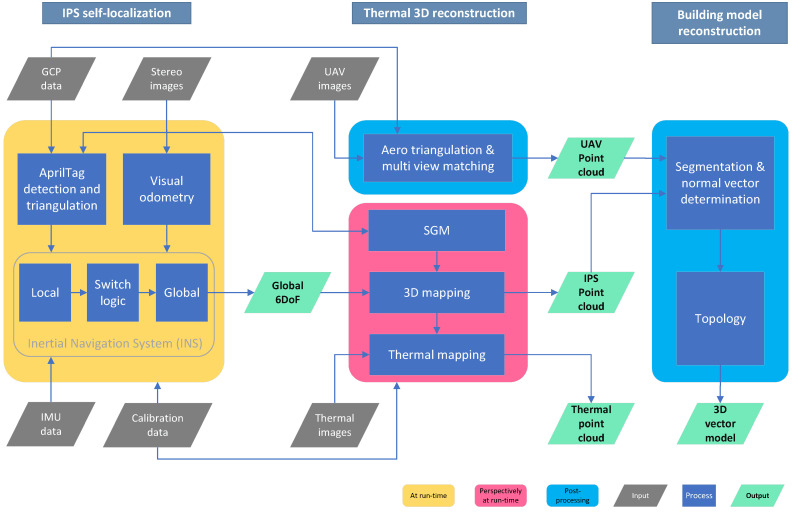
Workflow with our three thematic blocks, *IPS self-localization*, *Thermal 3D reconstruction*, and *Building model reconstruction*, structured according to processing time.

**Figure 6 sensors-22-04745-f006:**
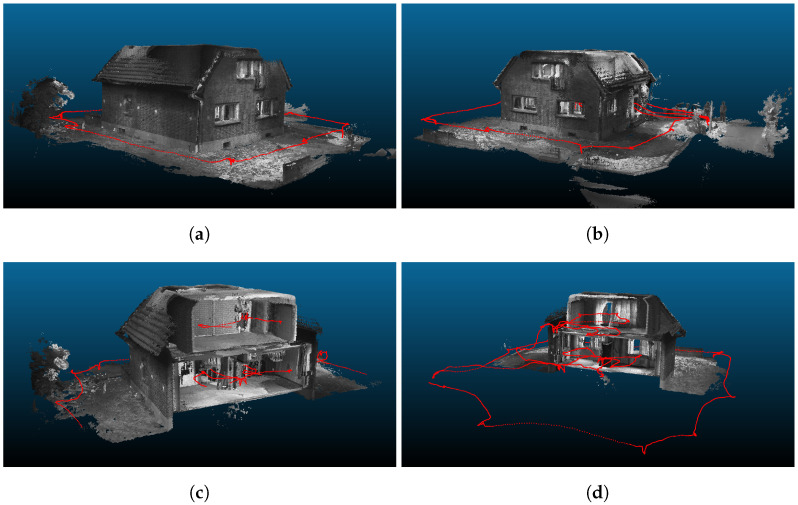
Multiple views of the point cloud generated from the IPS trajectory (red dots) and the IPS stereo images. (**a**,**b**) Building exterior views. (**c**,**d**) Interior view showing the two floors.

**Figure 7 sensors-22-04745-f007:**
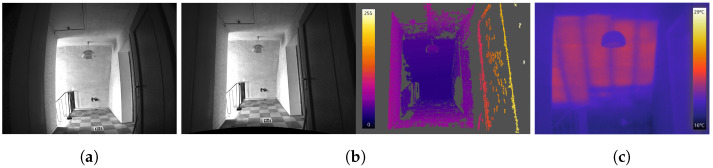
(**a**) Left IPS camera image. (**b**) Rectified left image and color coded disparity image generated by SGM (color bar with disparities values (gray values = no data)). (**c**) IR sensor image (color bar with temperature values).

**Figure 8 sensors-22-04745-f008:**
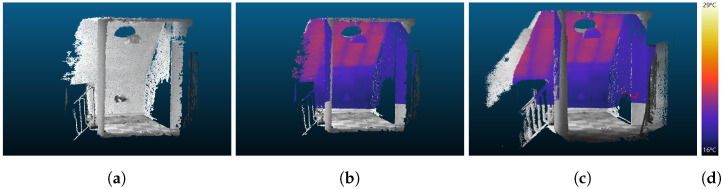
(**a**) Point cloud from one stereo image pair. (**b**) Values from one IR image assigned to the point cloud from one stereo pair recorded at the same time. (**c**) Values from one IR image are assigned to a point cloud accumulated from multiple images that have been recorded from the red dotted positions. Some parts (e.g., behind the ceiling lamp) are hidden in the IR camera view and are, therefore, excluded from IR mapping; (**d**) color bar with temperature values.

**Figure 9 sensors-22-04745-f009:**
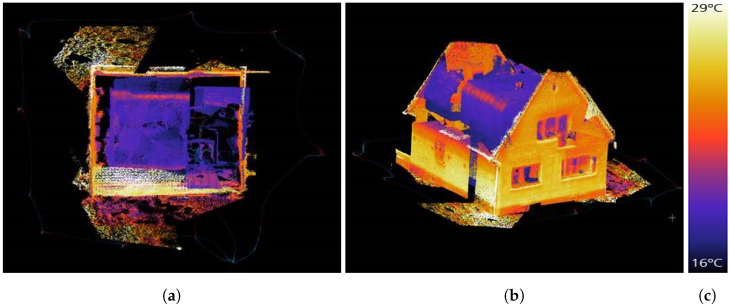
Complete IR mapped point cloud combining IPS and thermal camera data: (**a**) top view; (**b**) side view; (**c**) color bar with temperature values.

**Figure 10 sensors-22-04745-f010:**
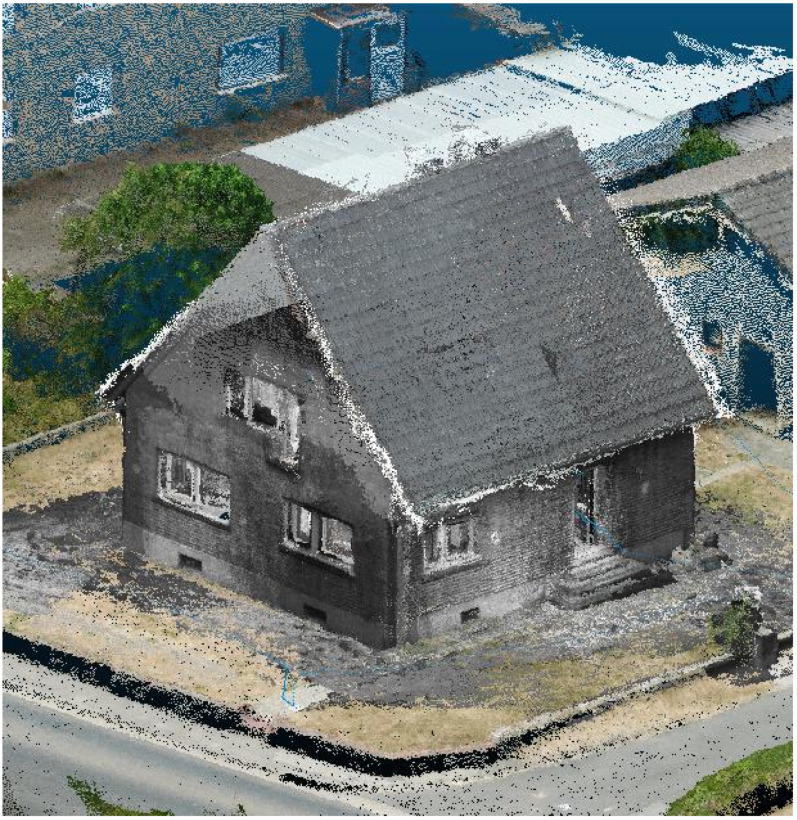
Complementary 3D reconstruction from UAV imagery (colored parts inclusive of the gray roof) combined with IPS on-ground derived point cloud (ground surroundings and building walls shown in grayscale).

**Figure 11 sensors-22-04745-f011:**
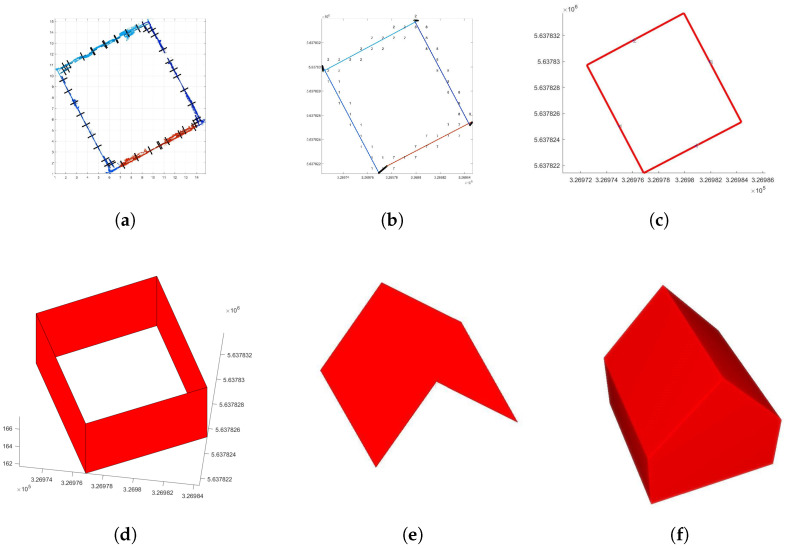
3D building vector model reconstruction: (**a**) point cloud voxel projected onto 2D grid; (**b**) calculated segments; (**c**) closed contour; (**d**) erected contour; (**e**) roof structures; (**f**) complete 3D vector model. The segmented ground plan parts in (**a**,**b**) are highlighted by random colors before accumulation in (**c**).

**Figure 12 sensors-22-04745-f012:**
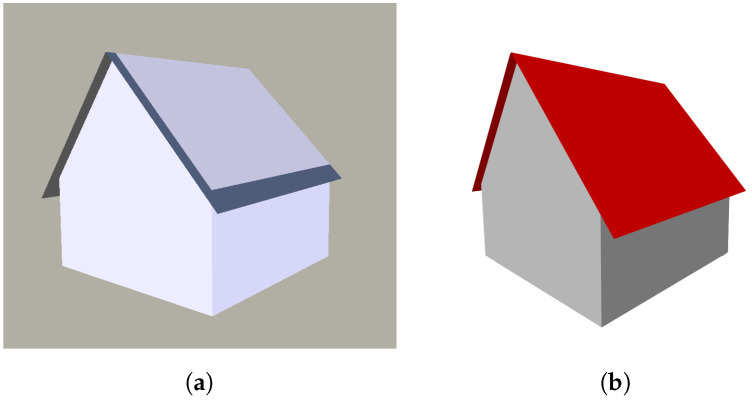
3D building reconstruction: (**a**) simplified model with roof overhang; (**b**) semantic model.

**Figure 13 sensors-22-04745-f013:**
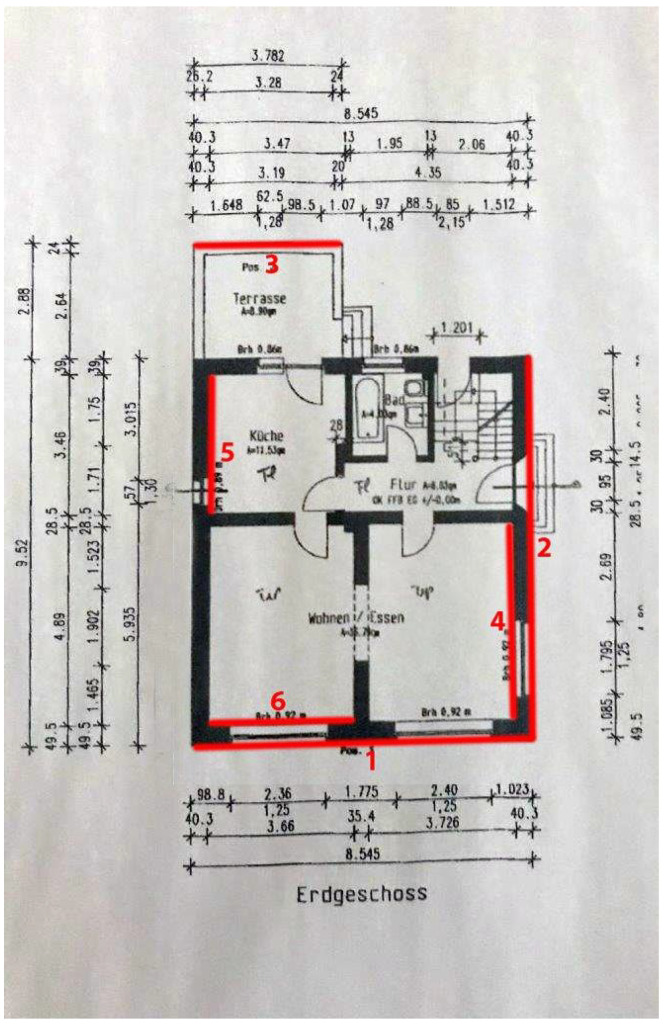
Floor plan with reference data and measured (red) wall parts.

**Figure 14 sensors-22-04745-f014:**
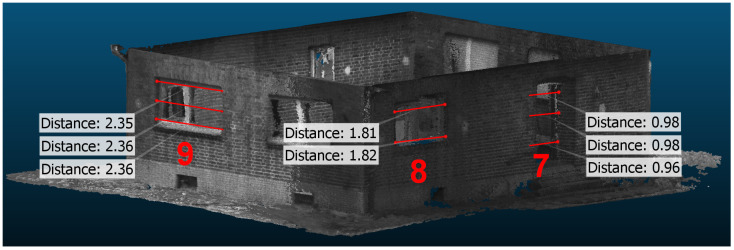
Subset of the IPS point cloud with three examples of distance measurements (m), see second column of Table 3—Ids 7, 8, and 9.

**Figure 15 sensors-22-04745-f015:**
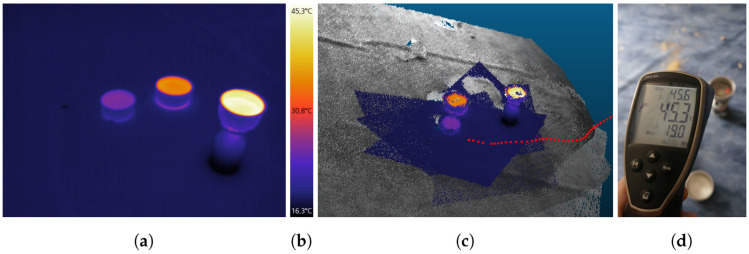
(**a**) Color-coded Optris PI 450 thermal image of water-filled pots. (**b**) Color bar with temperature values. (**c**) Partial 3D point cloud with trajectory (red dots). (**d**) Reference measurement with an infrared thermometer.

**Table 1 sensors-22-04745-t001:** Camera specifications of the integrated positioning system (IPS).

	Prosilica	Optris
Model	GC-1380H	PI 450
Camera	visible	thermal
Sensitivity (μm)	0.4–0.9	7.5–13
SNR or NETD	n/a	40 mK
Resolution (px)	1360×1024	382×288
Dynamic range (bit)	12	12
Pixel pitch (μm)	6.45	25
Focal length (mm)	4.8	10.5
Field of view (deg)	85	53
Frame rate (Hz)	10	27

**Table 2 sensors-22-04745-t002:** Calibration results for the trifocal sensor.

	IPS Left	IPS Right	Optris PI 450
c_k_ (px)	776.6	773.5	408.5
x_0_ (px)	711.2	681.5	188.1
y_0_ (px)	546.3	540.7	146.2
k_1_	−0.273	−0.257	−0.187
k_2_	0.168	0.118	−0.008
k_3_	−0.069	−0.029	0.434
t_x_ (cm)	-	−20.162	−0.314
t_y_ (cm)	-	−0.036	7.968
t_z_ (cm)	-	0.060	3.368
ω (rad)	-	0.0087	−0.0183
ϕ (rad)	-	0.0077	0.0231
κ (rad)	-	0.0048	0.0009

**Table 3 sensors-22-04745-t003:** Selected distances measured in the IPS point cloud and 3D vector model compared with the references shown in Figure 13.

Id (i)ndoor, (o)utdoor	Reference (m)	IPS Point Cloud (m)	Average Difference (m)	3D Vector Model (m)
1 (o)	8.55	8.54–8.58	0.01	8.46
2 (o)	9.52	9.47–9.51	0.03	9.38
3 (o)	3.78	3.66–3.70	0.10	-
4 (i)	4.89	4.86–4.90	0.01	-
5 (i)	3.46	3.38–3.42	0.06	-
6 (i)	3.66	3.78–3.82	0.14	-
7 (o)	0.95	0.96–0.98	0.02	-
8 (o)	1.80	1.81–1.82	0.02	-
9 (o)	2.36	2.35–2.36	0.05	-

**Table 4 sensors-22-04745-t004:** Object temperatures during the test run were measured by the thermal camera Optris PI 450. A non-contact infrared thermometer served as a reference.

Object	Reference (${}^{\circ }$C)	Optris PI 450 Measurements (${}^{\circ }$C)
Water Surface 1	31.4–31.7	30.0
Pot Wall 1	21.8–24.8	18.7–24.0
Water Surface 2	45.3–45.6	45.2
Pot Wall 2	22.0–27.6	22.4–26.1
Water Surface 3	27.1–27.6	25.8
Water Surface 4	35.7	35.3

## Data Availability

Not applicable.

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
