# Peer review of "Seamless Navigation, 3D Reconstruction, Thermographic and Semantic Mapping for Building Inspection"

_sensors, 2022, doi:10.3390/s22134745_

Round 1

Reviewer 1 Report

This paper presents an interesting study realizing a a workflow for seamless real-time navigation and 3D thermal mapping in combined indoor and outdoor environments in a global reference frame. This novel detection based method might overcomes some limitations by realizing seamless indoor and outdoor real-time self-localization with an integrated positioning system and combines this with thermal 3D mapping. In this way, it is possible to navigate between indoor and outdoor spaces and to quickly survey critical infrastructures. Although some technical problems exist in the current study, the authors have shown a very different but new opinion for seamless real-time navigation and 3D thermal mapping in combined indoor and outdoor environments in a global reference frame. Generally, this idea is practical and useful to perform a building inspection. However, the manuscript needs to be further revised according to the following comments before being accepted for publication.

My main concerns are about the performance of the target detection and recognition algorithm. It should be noticed that both images of the indoor/outdoor environments can be acquired by the complete IR mapped point cloud combining IPS and thermal camera data. Therefore, I wonder if these indoor images could achieve the same 3D accuracy, together with the images of the outdoor environments? For the IPS self-localization (as shown in figure 5), the measurement data of outdoor environments may easily be obtained. How about the indoor images? Will the performance degrades? More details should be given about that.

 Other comments:

1. Does the camera system calibration have to be executed every time before an inspection run?Whether to recalibrate when the reference frame changes?

2. How long will this process (shown in Figure 4) take to execute? How much a building's structural complexity affects it?

3. For the selection logic for switching from local to global navigation, what is its automatic identification accuracy? More explanations should be given.

4. A discussion on the appropriateness of the 3D model used in this study would be beneficial. Because different 3D buildings should have different structures, using the same model of the two floors for all buildings of one subjector would be sure to result in some inaccuracies.  

5. Was there any data augmentation? Whether there will be dislocation of different 3D data? If so, how to correct it? More algorithm details should be present in the paper.

Author Response

Dear Sir or Madam,

please find attached a revised version of the paper with blue marked text passages, which address Your suggestions and hopefully answer the questions.

The below listed answers were not included in the new version, as we don't see them necessarily included.

Q: Does the camera system calibration have to be executed
every time before an inspection run?Whether to recalibrate
when the reference frame changes?

A: In principle it is not necessary to calibrate the camera system before each inspection run. But as our thermal camera lens system has a slightly thermal walk and we wanted to be sure to have optimal results, we decided to calibrate before every inspection run. We mentioned this fact at the end of the calibration introduction.

Geometrical camera calibration and the reference frame are in principle independent from each other. It is not necessary to recalibrate when changing the reference frame, assumed that the geometrical camera calibration does not change during the inspection run, and there was not such a behavior to be seen in the measurement data.

Q: A discussion on the appropriateness of the 3D model used in
this study would be beneficial. Because different 3D
buildings should have different structures, using the same model
of the two floors for all buildings of one subjector would be sure
to result in some inaccuracies.

A: We don’t use the same model for different buildings. A digital model is always generated for every inspected building and for every particular part of it. We don’t use any a priori information or digital models to support our workflow.

Q: Was there any data augmentation? Whether there will be
dislocation of different 3D data? If so, how to correct it? More
algorithm details should be present in the paper.

A: No data augmentation was done or user and no additional data reference (e.g. a laser scan, whom to compare or match/ correct against)

Sincerely Yours,

The authors

Reviewer 2 Report

Excellent paper

Author Response

Dear Sir or Madam,

thank You very much for Your kind review. For Your Information please find attached a new paper version with some additional text passages and corrections, the other two evaluators had asked for.

Sincerely Yours,

The authors

Reviewer 3 Report

Dear Authors,

I would like to provide my review for your manuscript, sensors-1759035, in the attached PDF. I enjoyed reading your work, and I hope my review will be helpful for your research.

Thank you.

Author Response

Dear Sir or Madam,

please find attached a revised version of the paper with brown color marked text passages, which address your suggestions.

Sincerely Yours,

The authors
